# The Effect of Diets Containing High-Moisture Corn or Triticale Grain on Animal Performance and the Fatty Acid Composition of Lamb Muscles

**DOI:** 10.3390/ani12223130

**Published:** 2022-11-13

**Authors:** Cezary Purwin, Paulina Maria Opyd, Maja Baranowska, Marta Borsuk-Stanulewicz

**Affiliations:** Department of Animal Nutrition and Feed Science, University of Warmia and Mazury in Olsztyn, Oczapowskiego 5, 10-719 Olsztyn, Poland

**Keywords:** lamb, high-moisture corn, triticale, digestibility, fatty acid composition

## Abstract

**Simple Summary:**

The growing demand for functional foods has increased consumer interest in lamb and hogget. These types of meat deliver considerable health benefits; they are easily digestible, nutritious, and have desirable sensory attributes. In Poland, lamb meat is expensive due to high production costs and a high supply of lambs of wool breeds that are not suitable for fattening due to a slow growth rate. In view of the ongoing search for effective methods of producing high-quality lamb, the aim of this study was to compare the effect of diets containing high-moisture corn and triticale, as cheap grain feed, on nutrient digestibility and utilization, fattening performance, and the content of fatty acids in meat, and to investigate the rationale behind increasing the inclusion rate of cereal grain in finishing diets for lambs of a dual-purpose breed.

**Abstract:**

The aim of this study was to evaluate the effect of diets with different inclusion levels of high-moisture corn (HMC) and triticale grain (TG) on nutrient digestibility and utilization, the growth performance of lambs, and the fatty acid profile of the leg muscle. The experiment was performed on 24 young rams, divided into four groups based on grain type and inclusion rate (50% or 75% DM). The higher inclusion rate of cereal grain and a lower concentration of crude fiber (CF) in the feed ration decreased CF digestibility by 26% and 35% in diets containing HMC and TG, respectively. Diets containing HMC increased crude fat digestibility relative to animals receiving TG. Final body weight and average daily gain were highest in lambs fed a diet containing 50% HMC. Grain type had no effect on nitrogen (N) retention or the proportions of SFA and UFA in lamb meat. Diets containing HMC decreased total cholesterol levels in the leg muscle. HMC can be a valuable component of diets for growing lambs when included at up to 50% of the ration. An increase in the dietary inclusion rate of cereal grain to 75% can be recommended only in the case of TG.

## 1. Introduction

In Poland, sheep are raised mainly for lamb exports. These types of meat deliver considerable health benefits [1]; they are easily digestible, nutritious, and have desirable sensory attributes [2,3]. Lamb and hogget also contain intramuscular fat with a favorable fatty acid profile (content of PUFA about 17% of sum of fatty acid) [4], as well as protein with a high biological value [5]. High prices of lamb meat are one of the limiting factors for lamb production. Lamb meat is expensive due to high production costs and a high supply of lambs of wool breeds that are not suitable for fattening due to a slow growth rate [6]. The energy content of feed rations determines the growth rate of lambs and, consequently, age at slaughter, which significantly influences carcass quality, meat quality, and the profitability of lamb fattening [7]. In Poland, lambs are traditionally finished on hay or haylage, supplemented with limited amounts of barley or oats. Finishing diets based on cereal grains are not administered due to high cost. However, recent years have witnessed considerable changes in the structure of the Polish cereal market, with a rapid increase in triticale (TR) and corn production. In 2021, TG and corn accounted for 37% of total cereal production in Poland [8].

Corn is the most widely grown cereal crop around the world [9]. It is also one of the most popular cereal grains in livestock diets, and according to FAO [10], approximately 63% of the produced corn is used for animal diets. Corn has high starch content, low fiber content, and more fat content with a high proportion of unsaturated fatty acids (UFAs) than other cereals (fat content in corn 44.6 g/kg DM, wheat 19.5 g/kg DM, barley 25.0 g/kg DM, rye 16.2 g/kg DM, sorghum 35.3 g/kg DM) [11], so it is an excellent source of energy in livestock production [12]. Most animal diets contain dry grain, but drying is an expensive method of grain preservation [13]. Ensiling is an alternative and cheaper method of grain preservation. This technology supports the production of high-quality silage, and it minimizes nutrient loss [14,15]. High-moisture corn (HMC) is also characterized by a higher nutritional value and higher dry organic matter digestibility (DOMD) than dry grain [16,17]. According to Wilkerson et al. [18], the total-tract digestibility of HMC in cattle is approximately 9% higher in comparison with dry grain.

Triticale grain is also widely used in the diets of various animal species and livestock groups. Triticale is a hybrid of wheat and rye [19]. This crop species possesses traits that are intermediate between both parent species, but it has lower agronomic and soil requirements than wheat and a higher feed value than rye [20]. In 2019, Poland was the largest producer of TG in the world, and it supplied a third of the global TG output (4.5 million tons) [10]. Triticale is a rich source of protein, and the protein content of selected TG cultivars can reach 13% on a dry matter (DM) basis [19]. Triticale protein contains nearly all exogenous amino acids [21], and the energy value of TG is similar to that of wheat. Starch is the main source of energy, and its content in TG reaches 57–65% [22]. Triticale grain also contains fewer anti-nutritional factors than rye (content of pentose in triticale grain-8.7 g/kg DM, rye 21.1 g/kg DM) [23,24].

In view of the ongoing search for effective methods of producing high-quality lamb, the aim of this study was to compare the effect of diets containing HMC and TG on nutrient digestibility and utilization, fattening performance, and the content of fatty acids in meat, and to investigate the rationale behind increasing the inclusion rate of cereal grain in finishing diets for lambs of a dual-purpose breed. We hypothesized that the different content of carbohydrate and different content and composition of fat of the cereals used, as well as their different proportion in the diet of lambs, may increase the nutrient digestibility and utilization, fattening performance, and the content of fatty acids in meat.

## 2. Materials and Methods

### 2.1. Cereal Grain Treatments

A medium-early hybrid corn cultivar, PR39H32 PIONEER Co. (FAO 230), was used as HMC in the experiment. Corn was harvested on 15–18 October 2020 (average daily temperature: 8 °C, average daily rainfall: 1.55 mm) at full ripeness and DM content of 606.2 g/kg, and it was crushed in a roll mill crusher (ROMILL, Brno, Czech Republic). The theoretical cutting length of the cutting unit was 5 mm (56 knives on the cutting drum). Corn was ensiled and stored in a silage bag (AG BAG Co., St. Nazianz, WI, USA). After 85 days of storage, samples of HMC were collected and used for the preparation of experimental diets.

Triticale grain was purchased on the market.

### 2.2. Animals and Feeding

The animal study protocol was consistent with the European guidelines for the care and use of laboratory animals, and it was approved by the Institutional Animal Care and Use Committee in Olsztyn, Poland (permission No. 73/2021). The experiment was conducted in the animal research station of the Department of Animal Nutrition and Feed Science, University of Warmia and Mazury in Olsztyn. The experimental materials comprised 24 young single rams of the dual-purpose long-wool Kamieniec breed. The lambs were weaned at approximately 60 days of age. At the beginning of the experiment, the average body weight of animals was 25.4 kg (±2 kg). The lambs were kept in individual cages for 45 days, and the amounts of feed and leftovers served to animals was recorded. Diets for growing sheep were composed according to the DLG system [25] with the assumption of a daily gain of 300 g. The lambs were divided into four equal groups (six animals per group) based on grain type and inclusion rate: diet containing 50% of high-moisture corn (HMC50%), diet containing 75% of high-moisture corn (HMC75%), diet containing 50% of triticale (TG50%), and diet containing 75% of triticale (TG75%). The diets also contained soybean meal and meadow hay (with length adjusted to approx. 6–0 cm) supplemented with food grade sodium bicarbonate (10 g/animal/day) and a mineral-vitamin premix (920 g/animal/day); 1 kg contained Zn-2500 mg, Mn-3000 mg, Co-15 mg, I-50 mg, Se-3 mg, Ca-240 g, P-120 g, Na-60 g, Mg-65 g; vitamins: A-300,000 IU, D3-30,000 IU, E-1500 mg; antioxidants: BHA (E 320)-100 mg, BHT (E321)-100 mg, ethoxyquin (E324)-107 mg, citric acid (E330)-200 mg. The animals had free access to feed that was served twice daily: in the morning (7.00 a.m.) and in the afternoon (4.00 p.m.) (70:30), and to water (via automatic drinkers).

### 2.3. Nutrient Digestibility and Nitrogen Balance

After 25 days of feeding (fattening), nutrient digestibility (balance method) and nitrogen (N) balance were determined in vivo. All animals were placed in individual metabolism cages. After a five-day adaptation period, the amount of feed served to animals was recorded, and leftovers, feces and urine were collected for five days.

### 2.4. Sampling and Chemical Analysis

Feed samples were subjected to chemical analysis before the experiment and once during the experiment. Leftover samples were stored at a temperature of minus 25 °C. Thawed samples were dried in Binder dryers with forced air circulation at a temperature of 60 °C for 48 h, and they were ground (similarly to other diet components) in the Retsch 200 ball mill to a 1 mm particle size.

Feces excreted by each animal were collected daily, and 10% samples were taken and frozen. At the end of the experiment, the samples were thawed, mixed, and two average samples were prepared. The first (fresh) sample was analyzed for N content, and the other sample was partially dried at a temperature of 60 °C for 48 h. The content of key nutrients was determined in samples of feed, leftovers and partially dried feces (after grinding) by standard methods [26]. The DM content of feeds and orts was examined by drying the samples at 105 °C for 24 h; wet feces samples were analyzed by drying at 135 °C for 2 h after drying at 600 °C for 24 h. Wet feces were used for N analysis. The OM content was calculated as the difference between the content of DM and ash (ash content was determined by the AOAC method 923.03). Ether extract (EE) and Kjeldahl N were determined by AOAC methods (920.39 and 984.13, respectively). NDFom and ADFom (NDF assayed with heat-stable amylase, sodium sulfate and expressed excluding residual ash; ADF expressed excluding residual ash) were analyzed by the methods of Van Soest et al. [27]. The results were used to determine nutrient digestibility and to calculate the nutritional value of lamb diets. To determine N balance, urine volume was recorded in all animals, and 10% samples were collected daily in containers containing 6.0 N sulfuric acid solution to prevent the escape of NH_3_. The N content of each sample was determined by the Kjeldahl method (Kjeldahl N) in the FOSS KjeltecTM 8400 Analyzer Unit, 8420 Auto Sampler. Nitrogen retention and utilization were calculated.

Crude fiber and fiber fractions (neutral detergent fiber (NDF), acid detergent fiber (ADF), and acid detergent lignin (ADL)) were determined in feed components and leftovers by the Van Soest methods [27], using the ANKOM 220 Fiber Analyzer (HMC, meadow hay) and FOSS TECATOR Fibertec 2010 System (TG, soybean meal). Starch content was determined by an enzymatic method, as described by Faisant et al. [28].

High-moisture corn was analyzed to determine the content of lactic acid and volatile fatty acids (acetic, propionic, butyric, formic) and ethanol. Immediately after collection, fresh HMC samples were homogenized with a Bosch blender in deionized water at a 5:1 ratio, and were passed through soft filter paper (Filtrak No. 388). The filtrate was deproteinized with a 24% solution of metaphosphoric acid, and centrifuged (13,000× *g* rpm, for 7 min). The supernatant was analyzed to determine the concentrations of acids and ethanol by high-performance liquid chromatography (HPLC SHIMADZU). Separation was carried out using the Varian METACARB 67H column (Organic Acids Column), mobile phase-0.002 M solution of sulfuric acid in deionized water, flow rate-1 cm^3^-min, UV detector, wavelength-210 nm. External fatty acid and ethanol standards were supplied by SUPELCO, and the lactic acid standard-by FLUKA. The content of ammonia N in water extracts of HMC was determined by the Conway microdiffusion method, and pH was measured with an HI 8314C pH–meter (HANNA Instruments).

The nutritional value of diets was expressed as metabolizable energy (ME), according to the DLG system [25]:ME = 0.0312 × gdEE + 0.0136 × gdCF + 0.0147 × g (DOMD-dEE-DCF) + 0.00234 × gCP
where: ME—metabolizable energy (MJ), dEE—digestible ether extract, dCF—digestible crude fiber, DOMD—dry organic matter digestibility, CP—crude protein (N × 6.25).

### 2.5. Body Weight

The body weights of lams were determined at the beginning of the experiment and after 15, 25, 35 and 45 days of feeding. At the end of fattening (day 45), all animals were slaughtered (after 24 h of fasting). Carcasses were weighed, and dressing percentage (hot carcass weight/live weight, %) and carcass gain (initial body weight × dressing percentage, %) were calculated. The efficiency of DM, ME and CP per kg of body weight gain and per kg of carcass weight gain was also determined.

### 2.6. Blood Analysis

On the last day of the experiment, blood was subsequently collected from the vena zygomatica into heparinized tubes, centrifuged for 10 min at 380× *g* and 4 °C, and the obtained plasma was then frozen until analysis. Serum glucose and urea levels were determined in lambs by enzymatic methods with the use of glucose oxidase (glucose) and urease (urea), the Epoll 200 spectrophotometer (POLL Ltd., Warsaw, Poland), and diagnostic kits (glucose—Cormay, urea—Alpha Diagnostics).

### 2.7. Content of Fatty Acids in Feeds and in the Leg Muscle of Lambs

The concentrations of fatty acids were determined in fat extracted from the leg muscle of lambs (collected 24 h post mortem) and in feeds. Fatty acid methyl esters were obtained by fat methylation using the Peisker method (methanol: chloroform: concentrated sulfuric acid mixture, 100:100:1 v/v/v). Ampules were transferred to test tubes and were placed in a dryer at a temperature of 80–90 °C for 2–3 h. Hexane was then added to the test tubes to extract fatty acid methyl esters [25]. The separation process was carried out using the Varian CP-3800 gas chromatograph with a flame-ionization detector (FID) and a capillary column measuring 50 m × 0.25 mm × 0. 25 μm; detector temperature −250 °C, injector temperature −225 °C, column temperature −500 °C → 2000 °C; carrier gas-helium (flow rate-1.2 mL/min); sample size 1 μL. Fatty acid methyl esters were identified in the samples with the use of Sigma Aldrich reference standards. The analyzed fatty acids were divided into the following groups: saturated fatty acids (SFA), monounsaturated fatty acids (MUFA) and polyunsaturated fatty acids (PUFA), as well as n-3 PUFA, n-6 PUFA, unsaturated fatty acids (UFA), hypocholesterolemic fatty acids (DFA), and hypercholesterolemic fatty acids (OFA) in the leg muscle of lambs.

Total cholesterol levels in lipids extracted from the leg muscle were determined colorimetrically by the method proposed by Rhee et al. [29], which relies on enzymatic reactions with esterase and cholesterol oxidase, with the EPOLL-20 spectrophotometer at a wavelength of 520 nm. The Pointe Scientific cholesterol assay kit was used.

### 2.8. Statistical Analysis

The results were processed statistically by analysis of variance (ANOVA) for factorial designs. The significance of differences between group means was determined by Duncan’s test. The calculations were performed using Statistica 9.0 software (Soft Inc., New York, NY, USA).

## 3. Results and Discussion

The chemical composition of lamb diets is presented in Table 1. The DM content of HMC was determined at 586.3 g, with similar concentrations of starch and CF (including ADF and ADL), whereas aNDF content was twice as low as in TG. Fat content was three times higher in HMC than TG (46.7 vs. 15.5 g/kg DM), and CP content was only somewhat lower in HMC than TG (110.1 vs. 122.1 g/kg DM). High-moisture corn had a pH of 4.02; it was more abundant in lactic acid than TG, and contained trace amounts of ammonia N. These parameters point to the high quality of silage, which could be attributed to the optimal moisture content of ensiled grain [30,31]. Meadow hay had high CF content, where aNDF accounted for more than 69% (DM basis). The analyzed feeds differed in fatty acid composition. High-moisture corn had the lowest SFA content and the highest MUFA content compared with the remaining feed ingredients. High-moisture corn was also characterized by a very low proportion of α-linolenic acid (C18:3). Triticale grain was most abundant in PUFA, including linoleic acid (C18:2).

Lamb diets differed in DM content (Table 2), which was higher in diets containing TG, regardless of the inclusion rate. Dry matter concentration was lowest in a diet containing 75% HMC. Diets with a higher inclusion rate of cereal grain had higher starch content (by 47–48% on a DM basis), whereas their CF content (including NDF and ADF) was more than twice lower. Therefore, these diets had a higher concentration of ME (by 5.6% and 2.4% on a DM basis). Crude protein content per kg of diet DM was similar, but lowest in a diet containing 75% HMC. Crude fat content was more than twice lower in diets containing TG as the source of starch. The compared diets did not differ in OM concentration.

Dry matter, ME, and CP intake was only somewhat higher in lambs fed a diet containing 50% HMC (DM basis), regardless of the fattening stage (Table 3, Figure 1). A high dietary inclusion rate of HMC (75% on a DM basis) did not decrease DM intake by young rams (g/kg BW^0.75^) in comparison with lambs fed TG (Table 3). This observation testifies to the high quality of HMC which has a relatively high DM content and a high proportion of lactic acid, as well as a desirable taste and aroma. These attributes positively affect silage intake [32].

Nutrient digestibility coefficients were determined in digestibility trials. The source of dietary starch had no significant effect on CP digestibility (Table 3). An increase in the inclusion rate of cereal grain (from 50% to 75% on a DM basis) led to a minor decrease (approx. 4%) in CP digestibility. McKeown et al. [33] reported a decrease in CP digestibility in lambs receiving triticale-based dried distillers’ grains with solubles—DDGS (40% or 60% on a DM basis) relative to animals fed diets with a high content of barley grain. The type and inclusion levels of grain in the feed ration had a significant influence on the digestibility coefficients of CF. Crude fiber digestibility was higher in lambs fed diets containing HMC (*p* < 0.01; *p* < 0.05). The higher inclusion rate of cereal grain (75% on a DM basis) reduced the concentration of CF in lamb diets and decreased CF digestibility by 26% and 35% in diets containing HMC and TG, respectively. As a result, CF intake decreased (*p* < 0.01) relative to lambs fed diets containing 50% cereal grain. These results are in agreement with the observations made by Sauvant [34], who found that a high dietary content of starch that is rapidly degraded in the rumen can decrease CF digestibility. Other researchers [35,36] also noted that diets containing large amounts of cereal grain compromise CF digestibility. In the present study, diets containing HMC increased crude fat intake and significantly increased (*p* < 0.01) crude fat digestibility (by 17% and 26%, respectively) relative to animals receiving TG. According to Goulas et al. [37], crude fat digestibility is enhanced in the gastrointestinal tract of sheep fed diets with increased crude fat content. Similar observations were made in this study, where the digestibility coefficient of crude fat was highest in lambs fed a diet with the highest concentration of this nutrient (75% HMC), and it was significantly higher (*p* < 0.01; *p* < 0.05) than in other animals. Therefore, the intake of digestible fat was higher (*p* < 0.01) in lambs receiving HMC. Diets with high starch content induced a minor increase in OM digestibility, but the observed differences were not significant. 

An analysis of the N balance revealed that N retention (expressed per unit of metabolic body weight) was similar in the analyzed lambs (Table 3). Nitrogen retention as a percentage of N intake and N digested was highest in lambs fed a diet containing 75% HMC, and lowest in those fed a diet containing 50% TG (DM basis). It should be noted that N utilization was higher in lambs fed HMC than in those receiving TG. These observations were confirmed by serum urea levels (Table 3), which were higher (*p* < 0.01) in lambs fed TR. Grain type and starch content had no effect on serum glucose levels. 

No significant differences in the body weight of lambs were found between groups during the entire fattening period (days 1–45) (Table 4). However, the growth rates of young rams differed across fattening phases (*p* < 0.01; *p* < 0.05). Average daily gain was highest between days 26 and 35 in lambs fed diets containing HMC (50% and 75%) and 75% TG. In the initial phase of fattening (the first 15 days), a diet containing 75% HMC (DM basis) significantly decreased the growth rate of lambs (*p* < 0.01; *p* < 0.05). At the end of the 45-day experiment, body weight and average daily gain (205 g/day) were highest in lambs fed a diet containing 50% HMC. The above could probably be attributed to the highest intake of DM, ME, and CP in this group. Borowiec et al. [38] reported a lower growth rate (approx. 150 g/animal/day) in Polish long-wool sheep fed dry or high-moisture corn added to the feed ration composed of concentrate and meadow hay (46.62% and 46.32%, respectively). In the cited study, body weight gains did not differ between lambs fed dry or high-moisture corn, which is consistent with the findings of Almeida et al. [39]. In the present study, no significant differences were found in average daily gain, carcass dressing percentage, or carcass weight between groups (Table 4). Despite the fact that live weight at slaughter was lowest in lambs fed a diet containing 75% HMC, this group was characterized by the highest dressing percentage, which is an important determinant of carcass quality. In the above group, dressing percentage was around three percentage points higher than in the remaining animals. This parameter was highest in lambs receiving a diet with the higher inclusion rate of HMC (75%), probably due to a less developed gastrointestinal tract in these animals. This assumption was validated by a higher percentage of carcass weight in total body weight. Similar values of dressing percentage were reported by Borowiec et al. [38].

The intake of DM and ME per kg of body weight gain was highest in lambs fed a diet containing 75% HMC (*p* < 0.01; *p* < 0.05) (Table 4). Crude protein intake did not differ significantly between groups. The dietary source and content of starch had no significant effect on the intake of DM, ME, or CP per kg of carcass weight gain.

The leg is a primal cut in the lamb carcass, therefore the fatty acid composition and cholesterol levels were analyzed in fat extracted from the leg muscle (Table 5). The introduction of a large amount of grains rich in unsaturated fatty acids into the diet did not affect the content of SFA, MUFA and PUFA in the analyzed muscles. It is probably related to the enzymatic activity of the rumen microflora. Enzymes produced by rumen microorganisms are responsible for the isomerization and hydrolysis of lipids present in feed raw materials and the conversion of unsaturated fatty acids to saturated acids [40]. Palmitic acid (C16:0; 27.42–28.23%) and stearic acid (C18:0; 14.65–17.64%) were the dominant SFA. In the composition of MUFA and total UFA was predominant oleic acid (C18:1; approx. 90% and 78–81%, respectively). A similar relationship was found by Margetin et al. [4], who compared lamb grazing feeding with feeding based on large doses of concentrate. In animals whose diets contained 75% HMC (DM basis), the concentrations of PUFA, in particular n-3 PUFA (*p* < 0.01; *p* < 0.05) in the leg muscle were lower than in the remaining groups. At the same time, these meat samples were characterized by the highest CLA content (*p* < 0.01; *p* < 0.05). It should be noted that the analyzed samples contained low amounts of conjugated diene isomers of linoleic acid (0.26–0.39 g per 100 g of total fatty acids) which deliver health benefits. The concentrations of C18:0, C18:2, and C20:4 were higher in the leg muscle of lambs fed a diet containing 50% HMC. The n-6/n-3 PUFA ratio was 7.78 in the leg muscle of animals receiving TG, and it was somewhat higher (8.94 and 8.91) in lambs fed HMC. The optimal n-6/n-3 PUFA ratio is 1–2. According to Wood et al. [41], this parameter should not exceed 4. In the current study, in all leg meat samples, the content of the most nutritionally desirable n-3 PUFA was several times lower than the content of n-6 PUFA. High values of the n-6/n-3 PUFA ratio and a very low proportion of CLA in the fatty acid profile of the leg muscle can be attributed to nutrition. Diets that are based largely on cereal grain increase the content of n-6 PUFA in meat, whereas the concentration of n-3 PUFA remains relatively low [42]. The meat of grass-fed cattle and lambs has a more desirable flavor than the meat of animals whose diets are composed mainly of cereal grain. French et al. [43] confirmed the positive effect of a grass-based diet on the content of n-3 PUFA in beef. The content of SFA is much lower in mutton produced in organic farming systems than in conventional systems [4]. Fisher et al. [44] analyzed the meat of three sheep breeds (Welsh Mountain, Soays, and Suffolk) and observed a specific breed effect on the performance of animals fed forage and concentrate and on the quality of their meat.

It could be reasonably expected that feed rations containing corn, a rich source of fat and fatty acids, would increase the proportion of UFA in lamb meat. Such a relationship was observed by Oprządek and Oprządek [45] in lambs, and by Sosin-Bzducha et al. [31] in calves. However, in the present study, the type and inclusion rates of grain in feed ratios had no significant effect on the fatty acid profile of fat extracted from the leg muscle. The above can probably be attributed to considerable changes in the fatty acid profile of corn during ensiling, as well as the rapid biohydrogenation of UFA in the rumen of young rams. Borowiec et al. [38] reported certain changes in the fatty acid profile during ensiling of HMC. In comparison with dry corn, HMC was more abundant in SFA and MUFA, but less abundant in PUFA. The greatest differences were noted in the content of C16:0, C18:0, C18:1, and C18:2. The cited authors did not observe significant changes in the fatty acid profile (SFA, MUFA, PUFA, CLA) of the leg muscle in young rams fed diets containing dry corn or HMC. McKeown et al. [46] found no significant differences in the concentrations of MUFA and PUFA in the subcutaneous tail fat of lambs fed diets containing 20% (DM basis) of corn- or triticale-based DDGS. In lambs receiving triticale-based DDGS, subcutaneous tail fat was characterized by a higher content of CLA and a more desirable n-6/n-3 PUFA ratio (8.70 vs. 10.65). Similar relationships in the proportions of n-6 and n-3 PUFA were observed in the present study.

The current study also demonstrated that diets containing HMC, in particular at the higher inclusion rate (75%), decreased total cholesterol levels in the leg muscle (*p* < 0.01) relative to lambs fed TG (Table 5). The above could suggest that corn, which is a rich source of fat and UFA, was directly responsible for the reduction in cholesterol levels in the leg muscle. Dietary UFA significantly affects metabolism and decreases cholesterol levels in humans and in animal products [47,48]. Borowiec et al. [38] also reported lower total cholesterol levels in the leg muscles of lambs fed dry corn or HMC relative to animals receiving barley grain. In the present study, cholesterol levels, in particular in lambs fed HMC, were lower than those reported by Chizzolini et al. [49] (66–75 mg cholesterol in 100 g of fresh tissue).

## 4. Conclusions

The growing demand for functional foods has increased consumer interest in lamb and hogget. The factor which determines the slaughter value of young animals is the concentration of energy in the ration, which is closely related to the share of concentrated feed in the ration. Nutrient digestibility was higher in lambs fed diets containing HMC compared with TG. Animals that received HMC were characterized by higher N utilization, higher average daily gain, higher carcass dressing percentage and lower cholesterol levels in meat. However, high doses of this grain (75%) lowered the PUFA content in lamb meat. Despite this, this study demonstrated that HMC can be a valuable component of diets for growing lambs when included at up to 50% (DM basis) of the ration. An increase in the dietary inclusion rate of cereal grain from 50% to 75% can be recommended only in the case of TG.

## Figures and Tables

**Figure 1 animals-12-03130-f001:**
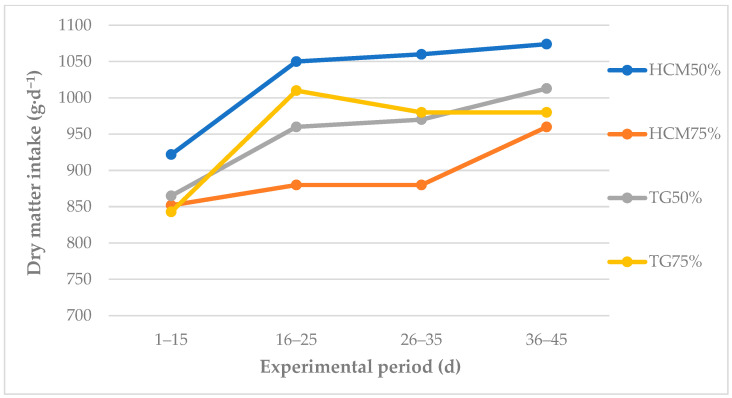
Average daily intake of dry matter (g·d^−1^) by lambs during the experimental period. HMC—high-moisture corn; TG—triticale grain.

**Table 1 animals-12-03130-t001:** Chemical composition of feeds.

Item	Feeds			
	High Moisture Corn (HMC) ^1^	Triticale Grain (TG)	Soybean Meal	Meadow Hay
Dry matter, g/kg^−1^ FM	586.3	886.9	880.5	854.7
Organic matter ^2^	983.8	978.8	934.6	964.9
Crude protein ^2^	110.1	122.1	467.7	141.0
Ether extract ^2^	46.7	15.5	11.5	11.5
Crude fiber ^2^	25.6	26.8	36.2	328.2
aNDF ^2^	80.3	185.8	83.8	695.4
ADF ^2^	31.5	35.0	59.2	358.2
ADL ^2^	13.5	15.2	14.2	43.9
Crude ash ^2^	16.2	21.2	65.4	35.1
Starch ^2^	618	647	69	-
Fatty acid profile (% of sums identified of acids)				
C_14:0_	0.06	0.22	0.17	1.35
C_15:0_	0.03	0.19	0.09	0.87
C_16:0_	15.09	19.61	15.41	29.45
C_16:1_	0.20	0.18	0.12	1.32
C_17:0_	0.12	0.13	0.15	0.97
C_17:1_	0.05	0.11	0.07	-
C_18:0_	2.18	1.68	3.98	4.05
C_18:1_	27.65	12.23	17.32	8.59
C_18:2_	52.84	57.59	52.99	28.65
C_18:3_	1.32	7.64	9.16	24.75
C_20:0_	0.46	0.25	0.34	-
C_20:1_	-	0.05	0.07	-
C_20:2_	-	-	0.05	-
SFA	17.94	22.21	20.19	36.69
UFA	82.06	77.79	79.81	63.31
MUFA	27.90	12.57	17.61	9.91
PUFA	54.17	65.23	62.20	53.40

^1^ in HMC (g/kg DM): lactic acid-8.16, acetic acid-3.89, butyric acid-1.69, propionic acid-0.58, formic acid-0.04, ethanol-1.33, ammonia-N-0.25 mg/kg DM, pH-4.02. ^2^ g/kg DM. ADF-acid detergent fibre; and-neutral detergent fibre assayed with heat stable amylose; DM-dry matter; FM-fresh matter; MUFA-monounsaturated fatty acids; PUFA-polyunsaturated fatty acids; SFA-saturated fatty acids; UFA-unsaturated fatty acids.

**Table 2 animals-12-03130-t002:** Ingredients (%DM) and chemical composition of the diet.

Item	Diets			
	HMC		TG	
	50%	75%	50%	75%
Ingredient				
HMC	50	75	-	-
TG	-	-	50	75
Soybean meal	15	15	15	15
Meadow hay	35	10	35	10
Chemical composition				
DM, g/kg^−1^ FM	725.1	656.0	875.5	881.2
OM ^1^	967.0	979.6	967.4	972.3
CP ^1^	176.0	169.2	180.9	178.3
EE ^1^	29.1	37.9	13.5	14.5
Starch ^1^	319.7	474.4	339.7	500.2
CF ^1^	131.8	55.3	130.5	55.4
aNDF ^1^	293.3	137.8	343.2	216.1
ADF ^1^	148.6	66.1	148.3	67.8
ME, MJ	11.67	12.32	11.38	11.66

^1^ g/kg DM. ADF—acid detergent fiber; aNDF—neutral detergent fiber assayed with heat stable amylose; CF—crude fiber; CP—crude protein; DM—dry matter; EE—ether extract; HMC—high moisture corn; ME—metabolizable energy; OM—organic matter; TG—triticale grain.

**Table 3 animals-12-03130-t003:** Nutrient intake and digestibility, nitrogen balance, and serum urea and glucose levels in lambs.

Item	Group				SEM	*p* Value		
	HMC		TG			Grain	Level	GxL
	50%	75%	50%	75%				
n	6	6	6	6				
Intake								
DM, g/day	1019	889	939	935	0.031	ns	ns	ns
g/LW^0.75^/day	76.1	71.4	71.8	73.1	0.015	ns	ns	ns
ME, MJ/day	11.91	10.95	10.68	10.98	0.390	ns	ns	ns
Nutrient digestibility, %								
CP	71.07	68.17	70.50	68.24	1.128	ns	ns	ns
EE	78.62	87.43 ^Aa^	60.96 ^B^	65.03 ^B^	2.575	<0.01	0.035	ns
CF	53.39	39.53 ^Ba^	46.74 ^AC^	30.54 ^Bb^	2.189	0.01	<0.01	ns
aNDF	61.40	45.29 ^Ba^	53.32 ^A^	35.02 ^Bb^	2.485	0.01	<0.01	ns
OM	77.04	79.20	76.56	77.68	0.82	ns	ns	ns
Digested, g/day								
OM	759	690	695	706	0.021	ns	ns	ns
CP	127	105	123	114	4.032	ns	<0.05	ns
CF	71 ^A^	19 ^B^	59 ^A^	16 ^B^		ns	<0.01	ns
aNDF	183 ^A^	55 ^B^	172 ^A^	71 ^B^	23	ns	<0.01	ns
Fat	24 ^A^	30 ^A^	8 ^B^	9 ^B^		<0.01	ns	ns
Nitrogen balance, g/day								
N intake	27.87	23.91 ^B^	29.99 ^A^	27.59	0.755	ns	<0.05	ns
N excreted								
in feces	8.08	7.02 ^b^	8.94 ^a^	7.76	0.398	ns	<0.05	ns
in urine	9.91	7.99 ^b^	11.37 ^a^	1021	0.599	ns	ns	ns
N digested	19.79 ^a^	16.89 ^b^	21.22 ^a^	19.83 ^a^	0.713	ns	<0.01	ns
N retention	9.88	8.90	9.68	9.62	0.566	ns	ns	ns
N retention/N intake, %	35.44	37.20	32.08	34.90	2.45	ns	ns	ns
N retention/N digested, %	49.91	52.69	45.59	48.51	3.24	ns	ns	ns
Plasma metabolites, mmol/l								
Urea-N	6.95 ^b^	6.39 ^b^	8.14 ^a^	8.28 ^a^	0.156	<0.01	<0.01	ns
Glucose	3.21	3.43	3.38	3.51	0.025	ns	ns	ns

Values not sharing the same superscript within a row are significantly different at *p* ≤ 0.05 (a,b) or *p* ≤ 0.01 (A,B,C). aNDF—neutral detergent fiber assayed with heat stable amylose; CF—crude fiber; CP—crude protein; DM—dry matter; EE—ether extract; HMC—high-moisture corn; LW—live weight; ME—metabolizable energy; N—nitrogen; OM—organic matter; SEM—standard error of the mean; TG—triticale grain.

**Table 4 animals-12-03130-t004:** Selected performance parameters in lambs.

Item	Diet				SEM	*p* Value		
	HMC		TG			Grain	Level	GxL
	50%	75%	50%	75%				
Body weight, kg								
Initial	25.83	25.00	25.75	25.03	0.791	ns	ns	ns
Final	35.08	32.25	33.67	33.50	0.800	ns	ns	ns
Average daily gain, g/day								
Days 1–15	161 ^a^	93 ^Bb^	182 ^A^	183 ^A^	11.79	0.009	ns	ns
Days 16–25	193 ^ac^	118 ^Bbd^	230 ^Aab^	142 ^Cd^	14.18	ns	0.02	ns
Days 26–35	298 ^A^	260 ^a^	170 ^Bb^	296 ^A^	17.11	ns	ns	0.01
Days 36–45	217 ^A^	230 ^A^	127 ^B^	183	13.02	0.006	ns	ns
Days 1–45	205	161	178	194	7.53	ns	ns	0.044
Carcass dressing percentage, %	46.57	49.43	46.11	46.17	0.95	ns	ns	ns
Carcass weight, kg	16.33	19.90	15.43	15.42	0.42	ns	ns	ns
Carcass gain, kg	12.00	12.30	11.59	11.51	0.39	ns	ns	ns
Efficiency								
DM, kg								
per kg of BW gain	4.92 ^b^	5.52 ^Aac^	5.36 ^a^	4.86 ^Bbc^	0.09	ns	ns	0.011
per kg of CW gain	3.79	3.27	3.56	3.67	0.08	ns	ns	ns
ME, MJ								
per kg of BW gain	57.28 ^b^	67.98 ^a^	61.14 ^b^	60.18 ^b^	1.94	ns	ns	ns
per kg of CW gain	44.16	40.00	40.47	42.61	1.40	ns	ns	ns
Crude protein, g								
per kg of BW gain	864	933	972	865	19.22	ns	ns	ns
per kg of CW gain	666	549	643	652	17.81	ns	ns	ns

Values not sharing the same superscript within a row are significantly different at *p* ≤ 0.05 (a,b,c,d) or *p* ≤ 0.01 (A,B,C). BW—body weight; CW—carcass weight; DM—dry matter; HMC—high-moisture corn; ME—metabolizable energy; SEM—standard error of the mean; TG—triticale grain.

**Table 5 animals-12-03130-t005:** Content of fatty acids (% of total fatty acids) and total cholesterol in the leg muscle of lambs.

Item	Diet				SEM	*p* Value		
	HMC		TG			Grain	Level	GxL
	50%	75%	50%	75%				
C10:0	0.28	0.28	0.26	0.26	0.008	ns	ns	ns
C12:0	0.68	0.72	0.59	0.59	0.027	0.034	ns	ns
C14:0	5.58	6.29 ^a^	5.33 ^b^	5.66	0.157	ns	ns	ns
C14:1	0.20	0.23	0.23	0.24	0.009	ns	ns	ns
C15:0	0.68	0.64	0.68	0.66	0.020	ns	ns	ns
C16:0	28.13	29.13 ^a^	27.42 ^b^	28.23	0.268	ns	ns	ns
C16:1	2.36 ^b^	2.47	2.64	2.70 ^a^	0.053	0.012	ns	ns
C17:0	1.28 ^B^	1.25 ^Bb^	1.43 ^a^	1.56 ^A^	0.038	<0.01	ns	ns
C17:1	1.01 ^C^	1.05 ^C^	1.21 ^B^	1.37 ^A^	0.035	<0.01	ns	ns
C18:0	17.64 ^Aa^	15.11 ^b^	16.29	14.65 ^B^	0.393	ns	0.005	ns
C18:1	35.50	37.51	37.68	37.85	0.452	ns	ns	ns
C18:2	4.17	3.42	3.87	3.84	0.124	ns	ns	ns
CLA	0.31 ^b^	0.39 ^Aa^	0.32 ^b^	0.26 ^B^	0.014	0.023	ns	0.005
C18:3	0.52 ^a^	0.44 ^Bb^	0.58 ^A^	0.57 ^A^	0.016	<0.01	ns	ns
C20:0	0.16	0.13	0.13	0.14	0.004	ns	ns	ns
C20:1	0.09 ^b^	0.09 ^b^	0.10 ^a^	0.11 ^a^	0.003	0.003	ns	ns
C20:2	0.04	0.03	0.04	0.04	0.002	ns	ns	ns
C20:4	1.14 ^a^	0.64 ^b^	0.97	1.04	0.078	ns	ns	ns
C20:5	0.04	0.03 ^b^	0.05	0.06 ^a^	0.005	0.011	ns	ns
C22:0	0.23	0.16	0.18	0.20	0.013	ns	ns	0.018
SFA	54.63	53.17	52.13	51.92	0.532	ns	ns	ns
MUFA	39.16	41.35	41.85	42.29	0.561	ns	ns	ns
PUFA	6.21 ^a^	4.95 ^b^	5.85	5.79	0.190	ns	ns	ns
n-3	0.55 ^a^	0.46 ^Bb^	0.63 ^A^	0.63 ^A^	0.020	0.01	ns	ns
n-6	5.34 ^a^	4.10 ^b^	4.90	4.90	0.200	ns	ns	ns
n-6/n-3	8.94	8.91	7.78	7.78	0.159	ns	ns	ns
UFA	45.37	46.30	47.69	48.08	0.544	ns	ns	ns
DFA	63.00	61.40 ^b^	63.98 ^a^	62.72	0.381	ns	ns	ns
OFA	37.00	38.60 ^a^	36.03 ^b^	37.28	0.358	ns	0.044	ns
Total cholesterol	62.53	58.16 ^B^	67.94	72.49 ^A^	1.920	0.007	ns	ns

Values not sharing the same superscript within a row are significantly different at *p* ≤ 0.05 (a,b) or *p* ≤ 0.01 (A,B,C). CLA—conjugated linoleic acid; HMC—high-moisture corn; MUFA—monounsaturated fatty acids; OFA—hypercholesterolemic fatty acids; PUFA—polyunsaturated fatty acids; SEM—standard error of the mean; SFA—saturated fatty acids; TG—triticale grain; UFA—unsaturated fatty acids.

## Data Availability

The data presented in this study are available from the corresponding author.

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
