# Peer review of "The Effect of Diets Containing High-Moisture Corn or Triticale Grain on Animal Performance and the Fatty Acid Composition of Lamb Muscles"

_animals, 2022, doi:10.3390/ani12223130_

Round 1
Author Response
Dear Timea Soos, Olsztyn, 09-11-2022
We thank Reviewers for the analysis and recommendations which helped us to improve the manuscript. We took special care to follow all of the detailed proofing marks throughout the text, and we hope that you find the result satisfactory. Please find the detailed response to Reviewers’ comments below.
Kind regards,
Paulina M. Opyd and co-authors
Reviewer #1:
Line 46-51 and 69-73: Introduction is not focused..
The introduction has been corrected. The first and fourth paragraphs have been merged with each other. The second indent has been redrafted.
Line 98: Please provide clarification. It is stated that lambs were divided into four groups. Then, in line 191, the authors mentioned that this experiment was a factorial design. There were two treatment groups (HMC and TR) and two levels of each group (50% and 75%). This will affect data presentation in all tables and explanations of results. So, a more clear description of the experiment design is needed.
The description of the research groups has been extended.
Line 99: What was the reference standard that diets were designed based on? for example, the National Research Council (NRC) or others?
Experimental diets were designed according to the DLG system.
In Chapter 2.2. Animals and feeding has been added the sentence: ‘Diets for growing sheep were composed according to the DLG system with the assumption of a daily gain of 300 g.’
Line 137: The procedure of determination of crude fiber of feeds or diets should be provided. Suggest removing crude fiber from table 1 and the text. The fiber fractions, including NDF, ADF and ADL are enough.
The procedure for determining the crude fiber has been added. We would prefer to leave the results for this ingredient in the article.
L191: Statistical analysis has to be clearly stated. Statistical significance was P < 0.05, and 0.05 ≤ P < 0.10 was considered a tendency to differ. Significantly difference at P < 0.01 is unnecessary. Besides, one of the P-values or * should be chosen to represent the significant differences among groups in Tables 3, 4 and 5; the error bar has to be shown in Figure 1.
While analyzing the text, we did not find the tendency indicated by the reviewer. We present the significance of the differences at the level of P <0.05 and P <0.01.
The designations of P in the tables have been unified.
The SEM value for the plotted values is so small that the error bars are invisible.
Line 225: The fatty acids composition of the complete diet should be presented in Table 2. And Tables 1 and 2 can be combined.
Tables 1 and 2 are so extensive that we have decided to present them separately.
Analysis of the lipid profile of the complete diet rations has not been performed. At the request of the reviewer, we can calculate the fatty acid content of the diet based on the lipid profile of the components.
The results part and discussion part should be addressed separately. Results and discussion have to be revised accordingly. This will improve readability and make sentences (and results) more clear.
Thank you for your comment, but when writing our publications, we prefer to present the results combined with a discussion. We will leave the decision about the necessity to separate chapters to the editor.
Performance and muscle fatty acids profile. Very poor discussion. Please elaborate.
The discussion’s logicality is unclear, e.g., "In turn" in Line 317. Authors should explain the turning relationship from the aspect of fatty acid metabolism. Similar problems should be carefully checked and rewritten in the relevant discussion section.
The discussion was expanded. The issue of isomerization and hydrolysis of lipids in the rumen was also discussed
Line 318-319: How can you state this? No significant effect on SFA was found…. Please revise all the similar mistakes, e.g., C16:0.
The sentence has been deleted.
Line 333-335: Why do the cereal grain-based diets largely increase n-6 PUFA and decrease n-3 PUFA in meat. What is machenism?
It is related to the large amount of n-6 acids and the small amount of n-3 acids contained in the grains. In the present study, we identified linoleic acid (C18: 2) as the major n-6 acid and α-linolenic acid (C18: 3) as the major n-3 acid in maize grain and triticale grain. Their content was as follows (% of sums identified of acids):
HCM – C18:2 – 52.84 C18:3 – 1.32
TG – C18:2 – 57.59 C18:3 – 7.64
I believe that the authors could improve the discussion if the FA composition and profile of the different feeds and diets were presented. Changes can be related to FA con of dietary treatments but not to forage alone, as concentrate composition varied among diets. Besides, the ratio of n-6 PUFA to n-3PUFA was much higher; why?
Table 1 presents the lipid profile of the feeds used.
Analysis of the lipid profile of complete diets has not been performed. At the request of the reviewer, we can calculate the fatty acid content of the diets based on the lipid profile of the components.
We have not presented the ratio of n-6/n-3 acids in the components and diets. However, the calculated n-6/n-3 ratio in HCM is 40 and in TG it is 7.53. These differences are related to the much lower content of n-3 acids in HCM.
Table 4: The initial and final body weights have to be listed in this table, for the authors have mentioned in line 281. ADG can also be listed in this table.
The initial and final body weights have been added to table 4. Table 4 already shows the average daily gain (ADG) for the entire experiment.
Line 359: Discuss why the 50% HMC had the highest C18:0 in comparison with the 75% HMC and 75% TG, which is related to de novo SFA synthesis. Most stearic acid present in tissues is formed by stearoyl-CoA desaturation of vaccenic acid formed in the rumen and is also desaturated in tissued to oleic acid.
In our opinion, it is related to the activity of ruminal microorganisms. Enzymes produced by rumen microorganisms are responsible for the isomerization and hydrolysis of lipids present in feed raw materials and the conversion of unsaturated fatty acids to saturated acids. This probably influenced the de novo synthesis of SFA. We raised this issue in the article.
Line 365: Is this true? A more desirable n-6/n-3 PUFA ratio (8.70 vs. 10.65). I can not find this data in any table.
We emphasize that these are the results quoted from the work of McKeown et al. Effects of corn- wheat- or triticale dry distillers grains with solubles on in vitro fermentation, growth performance and carcass traits of lambs. Can. J. Anim. Sci. 2010, 90 (1), 502 99-108
In the quoted article, they can be found in Table 6.
Line 378: Conclusions. Can this be concluded? The results do not support the text. Rephrased.
The requests have been expanded.
Minor comments:
- Line 20: The abbreviation of Triticale grain is TR in the abstract, defined as TG in tables, and TR in Figure 1. Authors should unify the abbreviation of Triticale grain. Similar problems should be noted in the full text.
Standardized throughout the text.
Line124-125 and elsewhere: The 105oC, 1350C, 600C mean oC?
Corrected throughout.
Line 142-154: It does not make sense to the aim of this study. Suggest delete.
The conducted analysis was aimed at determining the quality of the obtained silage. The maize grain was silaged by us, and poor quality silage could have a negative impact on feed intake and lamb production performance.
Line 168: When was the blood sample collected? Any available information should be provided.
Sentence added: „On the last day of the experiment, blood was subsequently collected from the vena zygomatica into heparinized tubes, centrifuged for 10 min at 380×g and 4°C, and the obtained plasma was then frozen until analysis.”
Line 184: Here and elsewhere, do not use the plural in abbreviations. FA stands for fatty acids; no plural is needed.
Corrected throughout.
Lines 201-207: High-moisture corn should be changed into HMC, for authors have already defined it. A similar problem is also suited for Crude fiber, and authors should check and correct those problems throughout the text.
We believe that the use of abbreviations and full names in discussions is interchangeable and makes the work easier to read. We don't see this as a mistake.
Replace “HCM” with “HMCh” in tables 2 and 3 and elsewhere.
Standardized throughout the text.
Line 313: Why did you use the leg muscle instead of the typical Longissimus thoracis muscle?
The leg muscle is the most frequently bought and prepared piece of lamb carcass in Poland, which is why we decided to conduct a test on it.
Table 5: The SEM of n-6/n-3 is missing; it has to be provided.
Added.

Reviewer 2 Report
The work may be published in the journal Animals, subject to the accompanying comments. Change the title and add a research hypothesis. The discussion lacked elements of the influence not only of the share of the analysed cereals (50 or 75%), but also their consequences on the chemical composition of the diet and nutritional value.
Detailed comments:
Line 3-4 - Change the title, don't use lambs twice,
Line 44 - Provide current data, e.g. for 2020 or 2021.
Line 50 - Specify the content of these ingredients or refer to other cereals.
Line 60 - Provide an appropriate source of literature.
Line 68 - State which anti-nutritional factors it applies to.
Line 72 - State which fatty acids are concerned.
Line 74 - Before the aim of the research, I did not find any research hypothesis, i.e. what the Authors wanted to achieve. Please specify and provide a hypothesis for these studies.
Line 82 - Complete the year of research and specify the climatic conditions, mainly rainfall and air temperature.
Line 104 - What proportions were the mentioned antioxidants used in?
Line 105-106 - Was the feed intake monitored, daily, weekly or throughout the study period?
Line 111 - How was urine and faeces collected and protected against changes in chemical composition?
Line 124 – better 135oC
Line 127 – AOAC 2000 - missing parentheses, correct
Line 161-162 - Were animals from the digestibility-balance test included in these weightings?
Line 169 - Enter the company and country of the manufacturer.
Line 172 - The Soxhlet method is concerned with the determination of the fat content of the feed, not with the determination of the fatty acid profile.
Table 1 - % of sums identified of acids
Line 196 - Please describe the research results separately and separate from the discussion.

Author Response
Dear Timea Soos, Olsztyn, 09-11-2022
We thank Reviewers for the analysis and recommendations which helped us to improve the manuscript. We took special care to follow all of the detailed proofing marks throughout the text, and we hope that you find the result satisfactory. Please find the detailed response to Reviewers’ comments below.
Kind regards,
Paulina M. Opyd and co-authors
Reviewer #2:
The discussion lacked elements of the influence not only of the share of the analysed cereals (50 or 75%), but also their consequences on the chemical composition of the diet and nutritional value.
The description of the influence of the analyzed cereals and their share on the chemical composition of the diets is described in the second paragraph of the discussion, and also presented in Table 2.
Moreover, the digestibility of the diets was determined (described in the fourth paragraph of the discussion and presented in Table 3), which shows the nutritional value of fodder.
Detailed comments:
Line 3-4 - Change the title, don't use lambs twice.
The title has been changed to: ‘The effect of diets containing high-moisture corn or triticale grain on animal performance and the fatty acid composition of lamb muscles.’
Line 44 - Provide current data, e.g. for 2020 or 2021.
Data corrected for 2021.
Line 48 – diet instead feed
Corrected.
Line 50 - Specify the content of these ingredients or refer to other cereals.
We added the fat content of various grains and the relevant literature.
Line 60 - Provide an appropriate source of literature.
Done.
Line 68 - State which anti-nutritional factors it applies to.
The phrase was added: “content of pentose in triticale grain - 8.7 g / kg DM, rye 21.1 g / kg DM” and the source of the literature was changed.
Line 72 - State which fatty acids are concerned?
We added the PUFA content in the lamb meat and the relevant literature.
Line 74 - Before the aim of the research, I did not find any research hypothesis, i.e. what the Authors wanted to achieve. Please specify and provide a hypothesis for these studies
Hypothesis added: ‘We hypothesized that the different content of carbohydrate and different content and composition of fat of the cereals used, as well as their different proportion in the diet of lambs, may increase the nutrient digestibility and utilization, fattening performance, and the content of fatty acids in meat.’
Line 82 - Complete the year of research and specify the climatic conditions, mainly rainfall and air temperature.
Meteorological data has been added.
Line 92 – station instead laboratory.
Done.
Line 104 - What proportions were the mentioned antioxidants used in?.
The content of antioxidants has been added in the methodology section.
Line 105-106 - Was the feed intake monitored, daily, weekly or throughout the study period?
The feed consumption was monitored daily.
Line 109 - in vivo in italics
Corrected.
Line 111 - How was urine and faeces collected and protected against changes in chemical composition?
The faeces were collected in faecal bags which were emptied twice a day in the morning and in the evening (7.00 and 19.00). 10% of the daily amount of faeces was collected and frozen.
The urine ran down a gutter placed under the grate floor of the balance cage into a container containing a 6N solution of sulfuric acid, after each diuresis the rest was rinsed with a small amount of water. Once a day, the collected urine was weighed and a 10% sample was collected into a sealed collecting vessel from which, after the end of the proper period, average samples were taken for analysis.
Line 124 – better 135oC.
Corrected throughout the document.
Line 127 – AOAC 2000 - missing parentheses, correct.
The year has been deleted.
Line 128 – NDFom and ADFom – what is it?.
NDF assayed with heat-stable amylase, sodium sulfate and expressed excluding residual ash
ADF expressed excluding residual ash.
The explanation is in parentheses.
Line 161-162 - Were animals from the digestibility-balance test included in these weightings?
All animals were transferred to metabolic cages on the 25th day of the experiment, so they were weighed on the designated days.
Line 169 - Enter the company and country of the manufacturer.
Corrected.
Line 172 - The Soxhlet method is concerned with the determination of the fat content of the feed, not with the determination of the fatty acid profile
Corrected.
Table 1 - % of sums identified of acids
Corrected.
Line 196 - Please describe the research results separately and separate from the discussion
Thank you for your comment, but when writing our publications, we prefer to present the results combined with a discussion. We will leave the decision about the necessity to separate chapters to the editor.
